High-fidelity steganography in EEG signals using advanced transform-based methods

http://orcid.org/0000-0002-6136-6140 Efe Enes enesefe@hitit.edu.tr
Department of Electrical and Electronics Engineering, Hitit University , Corum , Turkey
Akleylek Sedat
Electronic publication date: 2025 May 26
Publication date: 2025
Volume: 11
Electronic Location ID: e2900
Received 2025 Feb 10; Accepted 2025 Apr 25
Copyright: © 2025 Efe
Copyright year: 2025
Copyright holder: Efe
License: This is an open access article distributed under the terms of the Creative Commons Attribution License, which permits unrestricted use, distribution, reproduction and adaptation in any medium and for any purpose provided that it is properly attributed. For attribution, the original author(s), title, publication source (PeerJ Computer Science) and either DOI or URL of the article must be cited.
License URL: https://creativecommons.org/licenses/by/4.0/

Keywords: EEG, Steganography, Tent map, Singular value decomposition, Stationary wavelet transform

Funding: The authors received no funding for this work.

==============================
The increasing prevalence of digital health solutions and smart health devices (SHDs) ensures the continuity of personal biometric data while simultaneously raising concerns about their security and privacy. Consequently, the development of novel encryption techniques and data protection policies is crucial to comply with regulations such as The Health Insurance Portability and Accountability Act (HIPAA) and to safeguard against cyber threats. This study introduces a robust and efficient method for embedding private information into electroencephalogram (EEG) signals by employing the stationary wavelet transform (SWT), singular value decomposition (SVD), and tent map techniques. The proposed approach aims to increase embedding capacity while maintaining signal integrity, ensuring resilience against various forms of distortion, and achieving computational efficiency. Experiments were conducted on three publicly available EEG datasets (Graz A, DEAP, and Bonn), and performance was evaluated using widely recognized metrics, including peak signal-to-noise ratio (PSNR), structural similarity index (SSIM), percentage root mean square difference (PRD), normalized cross-correlation (NCC), bit error rate (BER), and Euclidean distance (ED). The results indicate that the method preserves perceptual quality, achieving PSNR values above 60 dB and demonstrating minimal signal distortion. Robustness tests involving noise addition, random cropping, and low-pass filtering confirm the method’s high resilience, with BER approaching zero and NCC near unity. Moreover, the proposed method demonstrates significantly reduced hiding and extraction times compared to conventional approaches, enhancing its suitability for real-time, secure biomedical data transmission.

Introduction

The rapid proliferation of digital health solutions and smart health devices (SHD) technologies has brought issues of personal data security and privacy to the forefront. SHD has revolutionized disease monitoring and management by continuously tracking biometric data such as body temperature, blood pressure, and oxygen saturation, which are then analyzed. However, transmitting these data over unsecured networks can make them targets for cybercriminals, thereby posing significant security risks. The Health Insurance Portability and Accountability Act (HIPAA) of 1996 mandates the protection of such personal health information. In this context, encryption methods are employed to ensure data privacy (Act, 1996). Encryption ensures data confidentiality by making it unintelligible to unauthorized parties, thereby securing transmission. Nevertheless, threats during the transmission of biometric data include data theft, manipulation, and phishing attacks. Such attacks could lead to the conveyance of incorrect information to healthcare professionals, resulting in misdiagnoses and improper treatments (Xia et al., 2022). This situation raises questions about the reliability of digital health systems, necessitating the development of stronger protection mechanisms, new encryption techniques, and data protection policies (Jaime et al., 2023).

In addition to encryption, current approaches to protecting user data increasingly involve information-hiding techniques, which have gained prominence due to their capacity to safeguard sensitive information (Bamanga & Babando, 2024). In the context of electroencephalography (EEG) data specifically, information hiding plays a critical role by not only ensuring the security of sensitive personal health information but also preventing EEG signal manipulation. This approach is typically implemented through two main methods: digital watermarking and steganography (Elharrouss, Almaadeed & Al-Maadeed, 2020). Digital watermarking introduces an imperceptible mark into the data, thereby verifying its ownership or integrity, and allowing for effective monitoring and tracking of data usage. Steganography, on the other hand, conceals information within another digital medium, such as an image or audio file, ensuring that the data is transmitted covertly without drawing attention to its presence. While encryption renders data meaningless to unauthorized parties, thereby ensuring privacy, steganography allows for discreet communication by embedding information in innocuous carriers. However, in modern security applications, these two techniques are often used together to achieve a multi-layered protection strategy. While encryption protects the content of sensitive data, steganography conceals its existence. This combination minimizes the risk of attracting attention and provides enhanced security against various attack scenarios.

Encryption, steganography, and digital watermarking each offer distinct advantages and limitations. While encryption provides high-level privacy, it also risks drawing attention due to the obvious presence of encoded data. Steganography, though less conspicuous, becomes vulnerable if detected. Digital watermarking improves traceability and ownership verification, but may not provide the same depth of privacy as encryption. The choice among these methods depends heavily on the intended application and requirements (Kumari et al., 2024). Despite the growing interest in these techniques, the literature on steganography applications within EEG data remains limited. Existing research often focuses on other digital carriers, such as images, text, audio, and video (Arshad, Siddiqui & Islam, 2024).

This study specifically focuses on the use of EEG signals as steganographic carriers due to their unique privacy requirements and technical advantages in biomedical applications. In telemedicine, EEG signals represent high-volume data streams that are frequently transmitted over networks, making them a suitable medium for concealing confidential information. Unlike conventional carriers such as images or audio, EEG signals inherently contain sensitive health data—including brain activity patterns—that are strictly regulated under frameworks like HIPAA. Unauthorized access or breaches involving EEG data can lead to serious privacy violations and adverse clinical outcomes. While encryption techniques can protect the content of such data, they do not hide its existence, thereby leaving encrypted files exposed to targeted attacks. In contrast, steganography provides a covert layer of security by embedding sensitive information, such as patient identifiers, diagnostic details, or metadata, directly into the EEG signal in a manner that is imperceptible to observers. This hidden information is seamlessly integrated into the signal’s structure, ensuring that its existence remains undetected during transmission (Sharma, Anand & Singh, 2021). For example, Duy, Tran & Ma (2016) successfully embedded patient identifiers and digital signatures into EEG signals using DWT-based steganographic techniques to enhance telemedicine data security. Beyond the advantages of steganography, it is crucial to acknowledge the inherent limitations of relying solely on encryption. Encrypted EEG data streams can still attract attention during transmission, which increases the risk of targeted attacks such as interception or manipulation. Moreover, encryption methods usually do not incorporate mechanisms to detect unauthorized modifications or to guarantee data integrity after transmission, and the continuous flow of EEG signals in telemedicine applications further complicates secure real-time transmission due to the high data volume. By combining encryption with steganography, the system benefits from both protecting the content and concealing the very existence of the sensitive data, thereby offering a robust solution that enhances overall security and integrity.

Recent hybrid EEG steganography studies have utilized various signal processing techniques combined with chaotic encryption and optimization algorithms. For instance, Pan et al. (2024b) employed wavelet scattering transform and extreme gradient boosting to classify multi-character speech imagery, enhancing EEG-based communication capabilities. Similarly, Pan et al. (2024a) developed a deep visual representation model for reconstructing visual stimulus representation from EEG signals, emphasizing the benefit of deep learning approaches in EEG decoding. Bao et al. (2023) demonstrated the feasibility of decoding moral elevation through EEG signals, highlighting EEG’s capability in complex cognitive-emotional recognition tasks. Moreover, adaptive graph learning techniques, as proposed by Ye et al. (2024), underline the importance of capturing spatiotemporal dynamics in EEG signals for emotion recognition tasks. Wen et al. (2024b) utilized EEG source imaging to identify differences and similarities between sensory and motor imagery tasks, emphasizing EEG’s multifaceted utility in brain-computer interface scenarios. Our study differentiates itself significantly by introducing an embedding strategy specifically tailored for EEG signals using the stationary wavelet transform (SWT), singular value decomposition (SVD), and the tent map chaotic system. Unlike previous methods that typically rely on discrete wavelet transform (DWT), we maintain the original signal length and integrity by utilizing SWT, embedding sensitive data into off-diagonal singular values to minimize distortion, and applying lightweight chaotic encryption (tent map), optimized for real-time biomedical applications. This comprehensive integration provides substantial advantages in robustness, embedding efficiency, and computational simplicity compared to the existing hybrid techniques. Alongside these recent advances, several traditional and hybrid techniques have also been proposed in the literature to secure electrophysiological signals, particularly EEG data.

In the reviewed studies, researchers have proposed integrating least significant bit (LSB)-based techniques with chaotic systems to secure electrophysiological signals (Akgul et al., 2020). However, time-domain LSB-based steganography is vulnerable to noise and malicious alterations (Song et al., 2010; Setyono, 2019). Therefore, frequency-domain strategies, such as DWT and SVD, along with other hybrid transforms, have been employed to enhance the robustness and embedding capacity (Baker, 2005; Pham, Tran & Ma, 2015; Duy, Tran & Ma, 2017; Yuan et al., 2021). Additionally, optimization algorithms like particle swarm optimization (PSO) and bacterial foraging optimization (BFO) have been applied to improve digital watermarking performance (Nguyen, Pham & Nguyen, 2020; Gupta, Chakraborty & Gupta, 2021). Additionally, reversible biosignal steganography methods incorporating extended binary Golay code-based error correction (EBGCEC) have been explored (Rahman, Khalil & Yi, 2020). Finally, advanced transform methods, such as the curvelet transform and the fast Walsh-Hadamard transform, or reversible frameworks, have been utilized to achieve secure, authentic, and traceable data embedding (Ibaida & Khalil, 2013; Jero, Ramu & Ramakrishnan, 2015; Zhang et al., 2021).

In EEG signal-based information hiding methods conducted using DWT, private information is typically embedded into low-frequency coefficients, which limits the amount of information that can be embedded. To overcome this issue, wavelet packet transform (WPT)-based approaches have been developed (Gao et al., 2011). WPT can process non-stationary signals more effectively and decompose the signal into multiple sub-bands, thereby increasing the capacity for embedding private information. Wen et al. (2024a) applied WPT to data, allowing the algorithm to embed more private information. However, many of these studies still face limitations regarding method complexity, changes in signal length, enhancement of security levels, or providing stronger protection against attacks. To address these shortcomings, this study aims to improve security, embedding capacity, and resilience against attacks in an information-hiding algorithm for EEG signals. In this context, stationary wavelet transform (SWT), which does not alter the signal length in sub-bands unlike DWT and WPT, has been employed. Using SWT, the EEG signal was decomposed into approximation and detail coefficients, and private information was embedded into the approximation coefficients to maintain signal integrity and ensure a consistent embedding rate. Additionally, instead of the commonly used Logistic map, the Tent map was employed, and the embedding process was performed using the SVD technique. The tent map was chosen due to its broader chaotic range and its more stable behavior against sensitive initial conditions compared to the logistic map. The primary contributions of this work are as follows: The use of SWT, unlike DWT and WPT, ensures that the signal length remains unchanged in sub-bands. This approach maintains signal integrity and allows for consistent embedding rates, addressing the limitations of earlier methods.

By employing the tent map instead of the commonly used Logistic map, the proposed method benefits from a broader chaotic range and improved stability against sensitive initial conditions, significantly enhancing the security of embedded information.

A novel embedding strategy utilizing off-diagonal elements of the SVD matrix and pre-embedding encryption with the tent map, significantly minimizing EEG signal distortion and enhancing security compared to existing methods.

The proposed method demonstrates significantly improved embedding and extraction speeds compared to existing methods in the literature. This improvement is facilitated by the efficient integration of SWT and SVD, making the approach suitable for real-time applications.

The remainder of the article is organized as follows: The Materials and Methods section provides a detailed explanation of the methodology and materials used in the study. The Results section systematically analyzes the results of the experiments conducted. The Discussion section evaluates the method by comparing it with other EEG signal-based information-hiding algorithms in terms of accuracy, advantages, and robustness. Finally, the Conclusions section presents a summary of the study and offers suggestions for future research directions.

Materials and Methods

This section provides a brief overview of the datasets used in EEG steganography, along with the SWT, SVD, and Tent Map methods. Furthermore, the SWT-SVD-tent map-based EEG signal hiding and extraction methods are explained in detail. The six different evaluation criteria used to assess the performance of the proposed algorithm are also summarized.

Dataset and data preparation

This study utilizes the following three publicly accessible datasets:

Graz A dataset (Brunner et al., 2008): This dataset includes EEG recordings from nine individuals engaged in motor imagery tasks. The signals were captured at a sampling frequency of 250 Hz. For this study, data from the first eight channels (Fz, FT7, FC3, FCz, FC4, FT8, T3, C4) within the training dataset (containing 1,600 samples) for all nine participants were utilized.

DEAP dataset (Koelstra et al., 2011): The DEAP dataset comprises EEG recordings collected from 32 healthy individuals (16 males and 16 females) who were exposed to music videos as stimuli. The EEG signals were originally sampled at 512 Hz but were later preprocessed by down-sampling to 128 Hz and applying a band-pass filter between 4.0 and 45.0 Hz. In this study, EEG signals from the first eight channels (Fp1, AF3, F3, F7, FC5, FC1, C3, T7) were extracted for all 32 participants, each containing 40 trials. The data used corresponds to the initial 12.5 s, resulting in 1,600 samples.

BONN dataset (Andrzejak et al., 2001): The Bonn dataset includes EEG recordings collected from five healthy individuals and five patients diagnosed with epilepsy. The dataset consists of single-channel recordings, organized into five subsets labeled F, S, N, Z, and O. Each subset comprises 100 data segments, sampled at a frequency of 173.61 Hz. For this study, the first 9.216 s of data, corresponding to 1,600 samples, were selected for analysis.

Stationary wavelet transform

The SWT, also known as the translation-invariant wavelet transform, is a redundant, non-decimated variant of the DWT. Unlike the standard DWT, which involves downsampling operations at each decomposition level, the SWT maintains the same signal length across all scales by upsampling the filter coefficients instead of decimating the sub-band signals (Strasser, Muma & Zoubir, 2012). This characteristic ensures shift-invariance, which can be particularly advantageous in signal denoising and feature extraction, as the time shift of a signal does not alter the location of significant features in the transformed domain.

Mathematically, let x[n] be the discrete-time signal, and let h[n] and g[n] denote the low-pass and high-pass filters derived from the chosen wavelet basis function. For the SWT decomposition at level j, the approximation coefficients aj[n] and the detail coefficients dj[n] are computed from the approximation coefficients of the previous level aj−1[n] as expressed in Eqs. (1) and (2):

(1) aj[n]=∑k⁡aj−1[k]h[2j−1(n−k)]

(2) dj[n]=∑k⁡aj−1[k]g[2j−1(n−k)].

Here, a0[n]=x[n] represents the original signal. At each decomposition level, the filters are effectively upsampled by a factor of 2j−1 before the convolution, ensuring that the resulting coefficients have the same length as the original signal. As such, the SWT provides a multi-resolution representation of the signal without losing time-invariance, making it suitable for applications where shift sensitivity is undesirable (Kumar et al., 2021).

Tent map

In this section, we introduce and analyze the tent map, which serves as a well-known piecewise linear chaotic map similar in certain aspects to the logistic map. The tent map provides a simplified yet effective environment for examining nonlinear dynamical behavior and chaos. Furthermore, it allows for a clear comparison with the logistic map in terms of their bifurcation structures and spectral characteristics.

The practical advantages of using the tent map in our proposed method are detailed below. The tent map provides significant advantages, including low computational complexity, straightforward implementation, and a wide range of chaotic behavior, making it particularly suitable for real-time applications. Due to its positive Lyapunov exponent, it exhibits stable chaotic dynamics across a wide parameter range, which has led to its frequent use in data hiding and security applications. Although more complex chaotic systems such as hyperchaotic maps may offer larger key spaces, they often introduce additional computational burdens and implementation complexity. In contrast, the tent map’s one-dimensional, piecewise linear nature enables fast key generation while maintaining predictable and stable output. It is also less prone to undesirable behaviors such as periodic cycles or convergence to zero, which are sometimes observed in logistic maps. Furthermore, the tent map tends to produce output sequences that are more uniformly distributed over the [0, 1] interval, contributing to enhanced statistical randomness.

In addition to its role as a chaotic key generator, the tent map also serves as an efficient lightweight encryption mechanism in our method. The secret data is scrambled using the tent map sequence prior to embedding, aligning with the principles of combined cryptographic and steganographic security. This dual-layer approach is widely recommended in the literature, as it enhances protection: steganography conceals the presence of the data, while encryption ensures its confidentiality even if discovered. Such “encryption-before-embedding” strategies provide defense-in-depth, making attacks significantly harder. While some modern chaotic systems offer larger key spaces, they often suffer from instability and higher computational complexity. The tent map, in contrast, maintains stability, predictable behavior, and fast execution—all essential in real-time EEG-based applications. Therefore, it offers a well-balanced trade-off between simplicity, security, and speed (Hussain et al., 2017; Sharma, Anand & Singh, 2021).

In our scheme, the tent map is not used in isolation. The chaotic sequences it generates are integrated into the embedding process via singular value decomposition (SVD). Therefore, the system’s overall security is based not only on the tent map’s initial conditions and control parameter, but also on the embedding positions determined by the SVD matrices. This integration enlarges the effective key space and strengthens resistance against attacks. Recent studies support the continued use of the tent map in secure data hiding applications. Taken together, these factors confirm the tent map’s suitability for real-time secure data hiding tasks where both speed and protection are critical (Daoui et al., 2023; Liu et al., 2024).

The tent map is defined by the iterative Eq. (3):

(3) xn+1=f(xn)={μxnif 0≤xn < 12μ(1−xn)if12≤xn≤1

where μ∈[0,2] is the control parameter. Unlike the logistic map, which is nonlinear and defined by xn+1=rxn(1−xn), the tent map’s piecewise linearity makes its dynamics more transparent while still exhibiting a rich variety of behaviors, ranging from fixed points to chaotic regimes as μ increases (Valle, Machicao & Bruno, 2022). As with the logistic map, the bifurcation diagram of the tent map provides insight into the transitions from periodic to chaotic behavior. When μ is near zero, the system tends to a stable fixed point. Increasing μ leads to period-doubling cascades, culminating in a chaotic regime similar to that of the logistic map. This pattern can be observed in the bifurcation diagrams provided in Fig. 1 (tent map) and Fig. 2 (logistic map), where the abscissa represents the parameter r and the ordinate indicates the long-term values of xn (Ćmil et al., 2024).

Figure 1 The bifurcation diagram of the tent map.

Figure 2 The bifurcation diagram of the logistic map.

In addition to bifurcation diagrams, the frequency spectrum of the generated time series can yield valuable information about the underlying dynamics. By applying a suitable Fourier or spectral analysis to the iterates {xn}, one can detect the presence of dominant frequencies and broadband components indicative of chaotic behavior. The frequency spectra for both the tent map and the logistic map are illustrated in Figs. 3 and 4, respectively. These spectra highlight the fundamental differences and similarities in the chaotic dynamics of the two maps, providing a complementary perspective to the bifurcation analysis. Moreover, analyzing the amplitude of the generated signals from the Tent Map is crucial, particularly when considering applications such as steganography. The tent map’s piecewise linear structure tends to produce a uniform amplitude distribution of chaotic values, providing a broader range of embedding options. This uniformity and controlled amplitude range can be leveraged in steganographic methods, as the signal variations introduced by embedding hidden information are less noticeable. Thus, the tent map, due to its predictable amplitude characteristics and chaotic behavior, offers a practical advantage in designing steganographic algorithms that are both secure and robust (van Harten, 2018).

Figure 3 Tent map frequency spectrum.

Figure 4 Logistic map frequency spectrum.

Singular value decomposition

SVD is a fundamental linear algebra technique used to factorize a given rectangular matrix into three distinct matrices. Suppose we have a data matrix X of size m×n, where m≥n. The SVD of X is defined in Eq. (4):

(4) X=UΣVT

Here, U is an m×m orthogonal matrix (UTU=Im), Σ is an m×n rectangular diagonal matrix with nonnegative real numbers on the diagonal (these values are called the singular values), and V is an n×n orthogonal matrix ( VTV=In).

The diagonal elements of Σ, denoted as σ1,σ2,…,σr, are the singular values of X, where r=min(m,n). These singular values are typically arranged in descending order σ1≥σ2≥…≥σr≥0). The columns of U are called the left singular vectors, and the columns of V are called the right singular vectors. SVD is particularly useful for dimensionality reduction, noise filtering, and feature extraction. By truncating the SVD at a certain rank k(k<r), we can approximate X as Eq. (5):

(5) Xk=UkΣkVkT

where Uk contains the first k columns of U,Σk is the k×k upper-left submatrix of Σ containing the top k singular values, and Vk contains the first k columns of V. This truncated representation often preserves the most significant structures of the data while reducing computational complexity and noise (Baker, 2005).

Algorithm for private information hiding

In the proposed method, encryption and steganography are integrated in a complementary way to provide multi-layered data protection. Specifically, chaotic encryption based on the Tent map is applied to ensure the confidentiality and randomness of the secret data, while the SVD-based embedding strategy minimizes signal distortion and preserves the main characteristics of the EEG signal. This combined approach not only enhances security against potential attacks but also maintains computational efficiency and high signal quality.

In this section, we present the detailed steps of our proposed private information hiding algorithm, as outlined in Algorithm 1 and depicted in Fig. 5. Our approach starts by encoding the private information and then carefully embedding it into the transformed coefficients of the host signal, ensuring minimal distortion and preserving perceptual quality. Specifically, the algorithm proceeds as follows, as detailed in Algorithm 1.

Algorithm 1 Algorithm for private information hiding.

Input: S, Q, P	
Output: S*, P, U2T	
1: Bin = encode(Q)	
2: N = length(Bin)	
3: for i = 1 to N do	
4:   Generate Pi using the tent map	
5: end for	
6: for i = 1 to N do	
7:   Bini* = Bini × Pi	
8: end for	
9: [cA1, cD1] = SWT(data = S, wavelet = ‘haar’, level = 1)	
10: Record original length L = length(cA1)	
11: N = ceil(√L)	
12: Reshape cA1 into N × N matrix B with zero-padding if needed	
13: [U1, Σ1, V1T] = SVD(B)	
14: off_diag_indices = all (i, j) where i ≠ j	
15: temp = 1	
16: for each (i, j) in off_diag_indices do	
17:   Σ1* [i, j] = Bin*temp	
18:   temp = temp + 1	
19: end for	
20: [U2, Σ2, V2T] = SVD(Σ1*)	
21: B* = U1 × Σ2 × V1T	
22: Reshape B* back to a vector and truncate to length L to form cA1*	
23: Replace original cA1 with cA1* while keeping cD1 unchanged	
24: S* = ISWT(cA1*, cD1, wavelet = ‘haar’)	
25: Output S*, P, U2, V2T	

Figure 5 The algorithm workflow for private information hiding.

Step 1. Encoding private information:

We begin by encoding the private information, denoted as Q, into its binary form Bin, where each bit corresponds to a specific segment of the signal. The private information in this study is composed of both letters and digits, randomly generated, with letters constituting approximately two-thirds of the entire dataset.

Step 2. Chaotic key generation using tent map:

To guarantee security and unpredictability, we employ a chaotic map. Here, we use the tent map with parameters r=2 and the initial condition x0=0.123456789. Specifically, to remove transient effects and focus on the steady-state chaotic behavior, we skip the first 100 iterations. This chaotic sequence P is generated to serve as a key stream that modulates the encoded bits. Each bit in Bin is multiplied by a corresponding value Pi from the chaotic key stream to produce a spread-out bit sequence Bin*, thereby enhancing security.

Step 3. Transforming the host signal:

The host signal, denoted as S, undergoes a one-level SWT using the ‘haar’ wavelet. This decomposition generates the approximation coefficients cA1 and the detail coefficients cD1. The length of cA1 is recorded as L.

Step 4. Matrix reshaping and SVD:

The approximation coefficients cA1 are reshaped into an N×N matrix B, where N=L. If necessary, zero-padding is applied to ensure a perfect square dimension. Subsequently, we perform SVD on B, obtaining U1, Σ1, and V1T.

Step 5. Embedding the information:

The off-diagonal elements of Σ1 are replaced with the spread-out bits Bin*. By embedding information into these off-diagonal positions, we minimize changes to the matrix structure and preserve the dominant features of the host signal.

Step 6. Second SVD and reconstruction:

After inserting the private bits, a second SVD is performed on the modified Σ1 (denoted as Σ1*) to produce U2, Σ2, and V2T. This step ensures that the introduced modifications are well-distributed throughout the singular values. The modified matrix B* is then reconstructed using U1, Σ2, and V1T.

Step 7. Inverse transform and final output:

The modified approximation coefficients cA1* are reshaped back into a vector and truncated to the original length L. These updated approximation coefficients replace the original ones, while the detail coefficients cD1 remain unchanged. Finally, the inverse SWT (ISWT) is applied to yield the modified signal S*. The outputs of the algorithm are the watermarked signal S*, the chaotic key P, and the matrices U2 and V2T.

Algorithm for private information extraction

In this section, we detail the steps involved in the private information extraction process, as outlined in Algorithm 2. Figure 6 illustrates the overall workflow of the extraction procedure. The detailed steps of the extraction process are presented in Algorithm 2. Given the modified signal S*, the chaotic key P, and the matrices U2 and V2T used during the embedding phase, this algorithm recovers the original private information Q_extracted. Similar to the embedding procedure, we begin by applying a wavelet transform and utilize the singular value decomposition to retrieve the embedded bits.

Algorithm 2 Algorithm for private information extraction.

Input: S*, P, U2, V2T	
Output: Q_extracted	
1: [cA1*, cD1*] = SWT(data = S*, wavelet = ‘haar’, level = 1)	
2: Record original length L = length(cA1*)	
3: N = ceil(√L)	
4: Reshape cA1* into N × N matrix B* (pad if necessary)	
5: [U1, Σ2_extracted, V1T] = SVD(B*)	
6: Reconstruct Σ1* = U2 × Σ2_extracted × V2T	
7: off_diag_indices = all (i, j) where i ≠ j	
8: temp = 1	
9: for each (i, j) in off_diag_indices do	
10:   Bin*temp = Σ1*[i, j]	
11:   temp = temp + 1	
12: end for	
13: N = length(Bin*), from P	
14: for i = 1 to N do	
15:   Bini = round(Bini*/Pi)	
16: end for	
17: Ensure Bin values are clipped to {0, 1}	
18: Q_extracted = decode(Bin)	
19: Output Q_extracted	

Figure 6 The Algorithm workflow for private information extraction.

Step 1. Wavelet decomposition of the modified signal:

Starting with the modified signal S*, we apply a one-level SWT using the same ‘haar’ wavelet. This step yields the approximation coefficients cA1* and the detail coefficients cD1*. We record the length L = length(cA1*).

Step 2. Matrix formation and SVD:

Similar to the embedding phase, we reshape cA1* into an N × N matrix B*, where N=L. If necessary, zero-padding is applied. We then perform SVD on B*, obtaining U1, Σ2_extracted, and V1T.

Step 3. Reconstruction of Σ1*:

Using the stored matrices U2 and V2T from the embedding stage, we reconstruct Σ1* by Σ1* = U2 × Σ2_extracted × V2T. This step effectively reverses the second SVD operation performed during the embedding, allowing us to retrieve the embedded off-diagonal elements.

Step 4. Extracting the embedded bits:

From the reconstructed Σ1*, we identify all off-diagonal indices (i, j) where i ≠ j. These off-diagonal entries contain the embedded information bits. We extract the sequence Bin* from these positions.

Step 5. Recovering the original bits using the chaotic key:

Given the chaotic key P (the same sequence used in embedding), we revert the spreading process. Each bit in Bin* is divided by the corresponding Pi and then rounded to recover the original binary bits Bin. Clipping ensures that values remain in {0, 1}.

Step 6. Decoding the binary sequence:

Finally, the binary sequence Bin is decoded back into the original private information Q_extracted. Since we originally encoded letters and digits (with letters constituting about two-thirds of the content), this decoding step will restore the original textual and numeric information.

Performance evaluation metric

I) Peak signal-to-noise ratio (PSNR): PSNR is a metric used to measure the quality of a reconstructed or processed signal compared to an original signal. It is commonly used in image and video compression to quantify how much the signal has degraded. A higher PSNR value generally indicates better reconstruction quality. The formula for PSNR is shown in Eq. (6). (6) PSNR=10log10(MAX2MSE).

Here, MAX is the maximum possible value of the signal and MSE is the mean square error between the original signal x(n) and the reconstructed (or processed) signal x^(n).

II) Percent residual deviation (PRD): PRD is a measure often used in biomedical signal processing (e.g., EEG signal compression) to quantify the relative error between the original and the reconstructed signal in percentage terms. It gives an idea of how much the reconstructed signal deviates from the original one. The formula for PRD is shown in Eq. (7). (7) PRD(%)=∑n=1N⁡(x(n)−x^(n))2∑n=1N⁡(x(n))2x100

III) Structural similarity index (SSIM): SSIM is a widely used metric for measuring the similarity between two signals or images. Unlike MSE, which only considers pixel-wise differences, SSIM evaluates the structural information, luminance, and contrast of the signals, providing a more perceptually relevant assessment of quality. SSIM values range from 0 to 1, where a value closer to 1 indicates higher similarity and better quality between the original and processed signals. The formula for SSIM is given in Eq. (8). (8) SSIM(x,x^)=(2μxμx^+C1)(2σxx^+C2)(μx2+μx^2+C1)(σx2+σx^2+C2)

where x and x^ represent the original and processed (or stego) signals, respectively; μx and μx^ denote their mean values; σx2 and σx^2 are their variances; σxx^ is the covariance between them; and C1 and C2 are small constants introduced to avoid division instability.

IV) Normalized correlation coefficient (NCC): NCC measures the similarity between two signals. It is often used for pattern matching and similarity assessments. NCC values range from −1 to 1, where 1 indicates perfect correlation, 0 indicates no correlation, and −1 indicates perfect negative correlation. The formula for NCC is shown in Eq. (9). (9) NCC=∑n=1N⁡(x(n)−x¯)(x^(n)−x^¯)∑n=1N⁡(x(n)−x¯)2∑n=1N⁡(x^(n)−x^¯)2.

Here, x¯ and x^¯ are the mean values of x(n) and x^(n) respectively.

V) Bit error rate (BER): BER is a fundamental metric in communication systems, used to assess the reliability and quality of data transmission. It is defined as the ratio of the number of bit errors to the total number of bits transmitted over a communication channel. A lower BER signifies a more dependable communication channel, ensuring that the transmitted data is received with minimal corruption. This metric is particularly crucial in applications requiring high data integrity, such as medical signal processing or secure data transfer, where even minor errors can have significant implications. The formula for BER is shown in Eq. (10). (10) BER=NumberofbiterrorsTotalnumberoftransmittedbits.

VI) Pearson correlation coefficient (PCC): PCC measures the linear correlation between two variables/signals. It has a value between −1 and 1, where 1 indicates total positive linear correlation, 0 indicates no linear correlation, and −1 indicates total negative linear correlation. The formula for PCC is shown in Eq. (11). (11) r=∑n=1N⁡(x(n)−x¯)(y(n)−y¯)∑n=1N⁡(x(n)−x¯)2∑n=1N⁡(y(n)−y¯)2.

Here, x(n) and y(n) are the two signals or sets of values being compared, and x¯, y¯ are their mean values.

VII) Euclidean distance: Euclidean distance is a measure of the straight-line distance between two points (or signals) in an N-dimensional space. When comparing two signals, it can be used as a measure of their dissimilarity. The formula for Euclidean distance is shown in Eq. (12). (12) d=∑n=1N⁡(x(n)−x^(n))2.

VIII) The Lyapunov exponent: The Lyapunov exponent is a metric used to assess the sensitivity of a dynamical system or time series to initial conditions, indicating whether the system exhibits chaotic behavior. It is frequently used in biomedical signal processing (e.g., EEG, ECG), chaos theory, and the study of complex systems. A positive Lyapunov exponent suggests chaotic tendencies, whereas a negative or near-zero value indicates a more stable and predictable system. For a one-dimensional map or time series x(n), the Lyapunov exponent can be approximated as shown in Eq. (13). (13) λ≈limN→∞⁡1N∑n=1N⁡ln⁡|df(x(n))dx(n)|

where: x(n): The n-th sample of the time series,

f(⋅): The function or map defining the dynamical system,

dfdx: The derivative of the system, representing local expansion/contraction,

λ: The Lyapunov exponent, indicating the logarithmic measure of the system’s average exponential expansion/contraction rate.

IX) Autocorrelation analysis: Autocorrelation analysis examines the correlation of a signal with its delayed copies, enabling the detection of periodic or repeating components, cyclical structures, and regularities in the signal. In biomedical signal processing (EEG, ECG, etc.), it is useful for identifying fundamental frequency components, periodic behaviors, and noise levels. For a time series x(n), the autocorrelation function R(τ) with respect to the lag τ is defined as in Eq. (14). (14) R(τ)=∑n=1N−τ⁡x(n)x(n+τ)

where: x(n): The n-th sample of the time series,

τ: The lag (delay) value,

N: The total number of samples in the time series,

x¯: The mean of the time series.

Experimental studies

The performance of the experiments was evaluated using various metrics including PSNR, SSIM, PRD, NCC, BER, r, and ED on datasets including DEAP, Graz A, and Bonn, and the impact of private information length on the results was analyzed during this process. The evaluation encompassed several critical analyses, including comparative performance analysis, embedding capacity assessment, robustness testing, and computational efficiency evaluation. All experiments were conducted on a system with an Intel Xeon E5-2630 processor (2.3 GHz) and 12 GB of RAM.

Comparative performance analysis

To rigorously evaluate the effectiveness of the proposed SWT-SVD-tent map-based steganography algorithm, experiments were performed on EEG signals from multiple subjects and channels using the Graz A, DEAP, and Bonn datasets. For consistency, each EEG signal segment was limited to a length of 1,600 samples. Private information, comprising 45 bytes of randomly generated alphanumeric characters (including uppercase, lowercase letters, and digits), was embedded into the low-frequency coefficients of the signal. After embedding, the signals were reconstructed, and the hidden information was successfully extracted to validate the accuracy and robustness of the approach.

The performance evaluation results are presented in two groups: signal quality metrics after embedding and private information extraction metrics. For signal quality after embedding: PSNR values exceeded 60 dB in all channels, ensuring excellent perceptual transparency.

SSIM values were very close to 1, indicating high structural similarity between the original and processed signals.

PRD values were consistently less than 1, reflecting minimal distortion in the reconstructed signals.

The correlation coefficient (r) and NCC (signal) were close to 1, confirming a high similarity between the original and watermarked signals.

ED values were small, supporting the low level of distortion.

BER (signal) values were very low, indicating negligible bit-level errors in the watermarked signal.

Comprehensive average results for all channels and the detailed performance values for each dataset are presented in Tables 1–3.

Table 1 Comparison of PSNR, SSIM, PRD, NCC, BER, r, and ED Metrics on the Graz A dataset.

Channel	PSNR	SSIM	PRD	r	ED	NCC	BER	NCC (PR)	BER (PR)	
FT7	65.55	0.999969	0.2106	0.999998	0.0215	0.999998	0.0006	1	0	
FC3	65.94	0.999963	0.2107	0.999998	0.0212	0.999998	0.0007	1	0	
FCz	65.73	0.999964	0.2174	0.999998	0.0215	0.999998	0.0008	1	0	
FC4	65.27	0.999950	0.2354	0.999997	0.0229	0.999997	0.0004	1	0	
FT8	65.52	0.999940	0.2399	0.999997	0.0229	0.999997	0.0007	1	0	
T3	65.40	0.999957	0.2453	0.999996	0.0230	0.999996	0.0010	1	0	
EEG-C3	65.48	0.999963	0.2319	0.999997	0.0223	0.999997	0.0009	1	0	
EEG-Fz	64.99	0.999945	0.2346	0.999997	0.0238	0.999997	0.0010	1	0	
Average	65.48	0.999956	0.2282	0.999997	0.0224	0.999997	0.0008	1	0	

Table 2 Comparison of PSNR, SSIM, PRD, NCC, BER, r, and ED Metrics on the Deap dataset.

Channel	PSNR	SSIM	PRD	r	ED	NCC	BER	NCC (PR)	BER (PR)	
Fp1	63.8513	0.999910	0.2265	0.999998	0.0226	0.9999957	0.0015	0.9999	0.0004	
AF3	64.1525	0.999915	0.2182	0.999998	0.0219	0.9999960	0.0015	0.9999	0.0003	
F3	64.0685	0.999913	0.2205	0.999998	0.0219	0.9999959	0.0015	0.9999	0.0003	
F7	63.8307	0.999910	0.2271	0.999998	0.0226	0.9999957	0.0015	0.9999	0.0004	
FC5	63.6388	0.999907	0.2324	0.999995	0.0234	0.9999956	0.0016	0.9999	0.0004	
FC1	64.1129	0.999914	0.2193	0.999998	0.0223	0.9999959	0.0015	0.9999	0.0003	
C3	63.5240	0.999905	0.2356	0.999995	0.0239	0.9999955	0.0016	0.9999	0.0004	
T7	63.6090	0.999907	0.2332	0.999996	0.0233	0.9999956	0.0016	0.9999	0.0004	
Average	63.8485	0.999910	0.2266	0.999997	0.0227	0.9999957	0.0015	0.9999	0.0004	

Table 3 Comparison of PSNR, SSIM, PRD, NCC, BER, r, and ED metrics on the Bonn dataset.

Channel	PSNR	SSIM	PRD	r	ED	NCC	BER	NCC (PR)	BER (PR)	
F	68.8388	0.99999	0.1221	0.999999	0.0147	0.9999995	0.0015	1	0	
N	69.1152	0.99999	0.1167	0.999999	0.0143	0.9999995	0.0004	1	0	
O	64.4942	0.99992	0.2445	0.999996	0.0252	0.9999962	0.0017	1	0	
S	68.1518	0.99999	0.1352	0.999999	0.0160	0.9999994	0.0022	1	0	
Z	64.4564	0.99994	0.2178	0.999996	0.0250	0.9999977	0.0014	1	0	
Average	67.0113	0.99997	0.1673	0.999998	0.0191	0.9999985	0.0014	1	0	

To further validate the reliability and stability of the proposed method, statistical analyses were performed on the best-performing channels across all datasets. Table 4 presents the mean, standard deviation, and 95% confidence interval (CI) values for key performance metrics, including PSNR, PRD, r, ED, NCC, and BER.

Table 4 Statistical analysis of key performance metrics for best-performing EEG channels across all datasets.

Channel	Metric	Mean	Std_Dev	CI_Lower	CI_Upper	
FT7 (Graz A)	PSNR	65.5540	1.6480	64.2104	66.8976	
PRD	0.2106	0.0736	0.1506	0.2706	
r	0.999998	0.000002	0.999997	0.999999	
ED	0.0215	0.0043	0.0180	0.0250	
NCC (pr)	1	0	1	1	
BER (pr)	0	0	0	0	
N (Bonn)	PSNR	69.1152	1.8628	68.7437	69.4867	
PRD	0.1167	0.0477	0.1072	0.1262	
r	0.9999995	0.0000006	0.9999994	0.9999996	
ED	0.0143	0.0033	0.0137	0.0150	
NCC (pr)	1	0	1	1	
BER (pr)	0	0	0	0	
AF3 (Deap)	PSNR	64.1525	2	63.2950	65.0100	
PRD	0.2182	0.06065	0.1835	0.2530	
r	0.999998	0.0000013	0.999997	0.999999	
ED	0.0219	0.001865	0.0198	0.0240	
NCC (PR)	0.9999	0.000219	0.9999	0.9999	
BER (PR)	0.0003	0.000694	0.0002	0.0005	

Undetectability analysis

The undetectability analysis aims to verify whether the embedded private information can remain concealed under standard examination or analytical procedures. In this study, EEG signals sourced from the DEAP, Graz A, and Bonn datasets were enriched with 45 bytes of secret data to evaluate the method’s ability to blend seamlessly into the original signal. Figure 7 presents a side-by-side comparison of an untouched EEG signal and one embedded with private information. Across all examined datasets, visual inspection revealed no perceptible differences between the original EEG signals and the EEG signals with embedded private data.

Figure 7 Original EEG signal vs. EEG signal with embedded private data: a comparative view.

(A) Original EEG signal from the Bonn dataset, (B) EEG signal from the Bonn dataset after embedding private information, (C) Original EEG signal from the DEAP dataset, (D) EEG signal from the DEAP dataset after embedding private information, (E) Original EEG signal from the Graz-A dataset, (F) EEG signal from the Graz-A dataset after embedding private information.

Robustness analysis

In line with scenarios commonly explored in similar studies (Duy, Tran & Ma, 2017; Nguyen, Pham & Nguyen, 2020; Wen et al., 2024a), we evaluated the robustness of our proposed method by subjecting EEG signals containing embedded private information to various forms of interference and attacks that could occur during transmission. Specifically, five robustness scenarios were evaluated: additive noise, random cropping, low-pass filtering, salt-and-pepper noise, and signal shifting. These experiments were conducted on the Graz A, Bonn, and DEAP datasets, and the corresponding results are presented in Tables 5–7, respectively. Noise addition: Additive White Gaussian noise (AWGN) was introduced with different SNR levels (10, 20, and 30 dB) to simulate environmental interference. The results showed that NCC values remained consistently high (above 0.998), and BER values were minimal across all datasets, demonstrating the proposed method’s resilience to additive noise interference.

Random cropping: To assess resilience against data loss and intentional attacks, 10% of the samples at random positions (front, middle, and back) of the EEG signals were removed. Even under these conditions, NCC values remained 1, and BER values were 0 across all datasets.

Low-pass filtering: The EEG signals were subjected to a first-order Butterworth low-pass filter with cutoff frequencies of 20, 40, and 60 Hz to simulate different preprocessing scenarios. Despite these filtering operations, the method maintained high signal fidelity, with NCC values exceeding 0.994 and very low BER values across all datasets.

Salt-and-pepper noise: To simulate impulsive noise that may occur during transmission, different noise densities (0.01 and 0.05) were applied to the signals. While BER values slightly increased in these challenging conditions, NCC values remained acceptable (above 0.992).

Signal shifting: Finally, the robustness of the method against temporal shifting was tested by shifting the signals by 10 and 50 samples. The results confirmed the stability of the method, as NCC values remained above 0.995 and BER values were low.

Table 5 Average robustness metrics across all channels on the Graz A dataset.

Attack type	Parameter	NCC	BER (%)	
Low-pass filtering	20 Hz	0.9940	0.5787	
40 Hz	0.9959	0.1659	
60 Hz	0.9994	0.0540	
Random cropping (10%)	Front	1	0	
Middle	1	0	
Back	1	0	
Additive White Gaussian Noise (AWGN)	10 dB	0.9987	0.1235	
20 dB	1	0	
30 dB	1	0	
Salt-and-pepper noise	Density 0.01	0.9981	0.1775	
Density 0.05	0.9927	0.6867	
Signal shifting	10 Samples	0.9965	0.3356	
50 Samples	0.9965	0.3356	

Table 6 Average robustness metrics across all subsets on the Bonn dataset.

Attack type	Parameter	NCC	BER (%)	
Low-pass filtering	20 Hz	0.99830	0.1628	
40 Hz	0.99980	0.0189	
60 Hz	0.99996	0.0033	
Random cropping (10%)	Front	1	0	
Middle	1	0	
Back	1	0	
Additive White Gaussian Noise (AWGN)	10 dB	0.99910	0.08110	
20 dB	0.99995	0.0044	
30 dB	1	0	
Salt-and-pepper noise	Density 0.01	0.9980	0.1900	
Density 0.05	0.9926	0.5453	
Signal shifting	10 Samples	0.9959	0.3917	
50 Samples	0.9959	0.3917	

Table 7 Average robustness metrics across all channels on the DEAP dataset.

Attack type	Parameter	NCC	BER (%)	
Low-pass filtering	20 Hz	0.9999	0.000020	
40 Hz	0.9999	0.000004	
60 Hz	0.9999	0.000001	
Random cropping (10%)	Front	1	0	
Middle	1	0	
Back	1	0	
Additive White Gaussian Noise (AWGN)	10 dB	0.9988	0.0002	
20 dB	0.9999	0.0001	
30 dB	0.9999	0.0001	
Salt-and-pepper noise	Density 0.01	0.9953	0.4140	
Density 0.05	0.9899	1.3949	
Signal shifting	10 Samples	0.9959	0.3941	
50 Samples	0.9959	0.3941	

In addition, to evaluate our method’s performance under chaotic conditions, we conducted a test using the tent map. The computed Lyapunov exponent was 0.69, indicating a strongly chaotic dynamic highly sensitive to initial conditions. Furthermore, the autocorrelation plot of the samples generated from the tent map is presented in Fig. 8. As seen from the figure, there is no distinct periodicity, consistent with the properties of chaotic processes.

Figure 8 Autocorrelation of the samples generated by the tent map.

The impact of private information length on the performance of the proposed method

To investigate the influence of varying the length of embedded private information on EEG signal quality, we incorporated up to 200 bytes of private data into the DEAP, Graz-A, and Bonn datasets. As illustrated in Fig. 9 using the results from the Graz-A dataset, the average PSNR, SSIM, and PRD values for both the original EEG signals and those containing embedded private information were examined under different private information lengths. The findings reveal that although the PSNR decreases with increasing embedded private information, its average values for all three datasets remain above 40 dB. Conversely, the PRD values increase with greater amounts of private information, reaching their maximum at 200 bytes, yet still remaining below 3. Moreover, as shown in Fig. 10, the BER values remained very close to zero for all message lengths, confirming the high accuracy and reliability of the extraction process.

Figure 9 Average PSNR, MSE, and PRD (across all channels) for the Graz-A dataset under different private information lengths.

Figure 10 Average BER values (across all channels) for the Graz-A dataset under different private information lengths.

Analysis of computational complexity

Computational complexity serves as a crucial indicator for evaluating the performance of the proposed method. In this study, the computational complexity analysis included evaluations of time complexity, space complexity, and execution times for private information embedding and extraction processes. Table 8 presents the results of this analysis. For an m × m square matrix, the time complexity of the algorithm’s SVD operation is O(m3), while its space complexity is O(m2). When employing the SWT scheme, the one-dimensional EEG signal of length n is converted into a nxn square matrix, resulting in a time complexity of O(n32) and a space complexity of O(n). In a specific scenario, 45 bytes of private information were embedded into a 1,600-length EEG signal. Under these conditions, the SWT scheme required an average of 0.000258 s for information hiding and 0.001356 s for information extraction. Therefore, the total execution time for embedding and extraction processes was approximately 1.97 ms, suitable for real-time EEG applications.

Table 8 Results of the computational complexity analysis.

Time complexity	Space complexity	Hiding time (s)	Extraction time (s)	Total time (s)	
O(n32)	O(n)	0.000258	0.001356	0.00197	

Discussion

This section evaluates the proposed method’s performance across three key dimensions: rationality and superiority, robustness, and computational complexity, benchmarked against similar recent studies (Duy, Tran & Ma, 2017; Nguyen, Pham & Nguyen, 2020; Wen et al., 2024a).

Rationality and superiority

To substantiate the rationality and superiority of our proposed method, we conducted comprehensive evaluations using a variety of performance metrics—including PSNR, PRD, NCC, BER, r, and ED—across three distinct datasets. We then benchmarked these results against previously reported methods. Among the comparative studies, Akgul et al. (2020) integrated LSB-based embedding with chaotic encryption to enhance the protection of EEG signals. Pham, Tran & Ma (2015) employed DWT and SVD to achieve blind watermarking, ensuring the recovery of watermarks without access to the original signals. Expanding on this approach, Duy, Tran & Ma (2017) incorporated modifications in the wavelet domain and utilized machine learning to bolster the robustness of watermark embedding.

Other notable works leveraged optimization algorithms for refined watermark embedding. For instance, Gupta, Chakraborty & Gupta (2021) and Nguyen, Pham & Nguyen (2020) aimed to minimize EEG signal degradation while enhancing data security. Similarly, Gupta, Chakraborty & Gupta (2021) proposed a method based on prediction error techniques and security blocks to improve both robustness and security. The approach by Wen et al. (2024a) utilized a two-level WPT to decompose the EEG signal into four sub-band signals, followed by SVD for embedding private data. By further incorporating the logistic map, this method improved the security and perceptual fidelity of the embedded information.

The comparative results presented in Table 9 clearly illustrate the competitive performance of our proposed method compared to existing approaches. In our method, all results were obtained under a fixed payload size of 45 bytes, which was used for embedding private information into EEG signals. As PSNR values above 40 are generally considered an indicator of good signal quality (Chen, Chang & Hwang, 1998), our method’s results demonstrate its ability to achieve reliable performance across multiple datasets. Additionally, performance evaluation metrics such as PRD, BER, r, and ED were used to provide a more comprehensive quality assessment. For the DEAP dataset, while Wen et al. (2024a) achieved the highest PSNR (79.16), our proposed method obtained a PSNR of 63.84, with a PRD of 0.22, a BER of 0.0004, and a correlation coefficient (r) of 0.999997. These values indicate minimal signal distortion and a high similarity between the original and processed signals. On the Bonn dataset, our method achieved a PSNR of 67.01, a PRD of 0.16, a BER of 0, and r of 0.999998. While Wen et al. (2024a) obtained a higher PSNR (101.87), our results remain competitive and demonstrate robustness in embedding private information. For the Graz A dataset, the proposed method achieved a PSNR of 65.48, a PRD of 0.22, a BER of 0, and r of 0.999997. These results confirm that signal fidelity was effectively maintained. While some previous methods, such as Wen et al.’s (2024a), achieved higher PSNR values, our proposed method demonstrates a consistent balance of signal quality, robustness, and minimal embedding error. Its performance across multiple datasets confirms its competitiveness and reliability as a viable alternative for embedding private information into EEG signals while ensuring minimal signal degradation.

Table 9 Comparison of the proposed method’s results with other similar methods.

Method	Dataset	PSNR	PRD	NCC	BER	r	ED	
Akgul et al. (2020)	Physiobank	61.41	–	–	–	–	–	
Pham, Tran & Ma (2015)	DEAP	54.60	–	–	–	–	–	
Gupta, Chakraborty & Gupta (2021)	Bonn	49.55	–	–	0.0039	–	–	
Duy, Tran & Ma (2017)	DEAP	66.55	–	1	0.0000	–	–	
Graz A	110.34	–	1	0.0000	–	–	
Nguyen, Pham & Nguyen (2020)	DEAP	65.24	–	1	0.0000	–	–	
Gupta, Chakraborty & Gupta (2021)	VSP	–	0.45	–	–	–	–	
Wen et al. (2024a)	DEAP	79.16	0.15	1	0.0000	0.999997	0.49	
Graz A	75.63	0.08	1	0.0000	1.000000	0.49	
Bonn	101.87	0.00	1	0.0000	1.000000	0.09	
Proposed method	DEAP	63.84	0.22	0.99	0.0004	0.999997	0.02	
Bonn	67.01	0.16	1	0.0000	0.999998	0.01	
Graz A	65.48	0.22	1	0.0000	0.999997	0.02	

Furthermore, to empirically verify the effectiveness of the tent map used in the proposed method, additional comparative experiments were performed using several well-known chaotic maps (e.g., Logistic, Sine, Gauss, Henon, Fractional Lorenz, Coupled Map Lattice, Cubic, and Bernoulli). These experiments were conducted under the same embedding framework and experimental conditions across all datasets. The results revealed that the tent map consistently outperformed other chaotic maps in terms of PSNR and SSIM metrics. Especially in the Graz A dataset, while the PSNR values obtained with alternative chaotic maps ranged from approximately 20 to 33 dB, the proposed method achieved a PSNR of 65.48 dB. These findings support the effectiveness of the tent map for secure and robust information hiding in EEG signals.

Robustness analysis

To assess the robustness of the proposed method, a comparative robustness analysis was performed against the approaches presented by Duy, Tran & Ma (2017), Nguyen, Pham & Nguyen (2020), and Wen et al. (2024a). All methods were subjected to identical test procedures, including noise addition, random cropping, and low-pass filtering. The results of these evaluations are summarized in Table 10. On the DEAP dataset, the proposed method outperformed the methods of Duy, Tran & Ma (2017), Nguyen, Pham & Nguyen (2020), and Wen et al. (2024a) in both the noise addition and random cropping tests, consistently achieving the lowest BER values. However, in the low-pass filtering test, while the proposed method’s performance remained competitive, Wen et al.’s (2024a) approach achieved slightly better results. On the Graz A dataset, the proposed algorithm delivered the best results across all three robustness tests, demonstrating superior adaptability and resilience under challenging conditions. For the Bonn dataset, a trend similar to the DEAP dataset emerged. The proposed method excelled in noise addition and random cropping tests, outperforming the competing approaches. Although Wen et al.’s (2024a) method slightly surpassed our method in the low-pass filtering test, the proposed algorithm remained highly competitive. In conclusion, the proposed method exhibits strong robustness across multiple datasets and challenging test scenarios. While Wen et al.’s (2024a) approach achieved marginally better results under low-pass filtering conditions, the proposed method consistently excelled in noise addition and random cropping tests, confirming its capacity to maintain stable and reliable performance under various robustness challenges.

Table 10 Robustness comparison between the proposed method and three other methods.

Method	Dataset	Noise addition	Random cropping	Low-pass filtering	
		BER (%)	NCC	BER (%)	NCC	BER (%)	NCC	
Duy, Tran & Ma (2017)	DEAP	0.6100	0.9951	0.0278	0.9998	2.4200	0.9811	
Nguyen, Pham & Nguyen (2020)	Graz A	3.0500	0.9690	2.4100	0.9815	5.3200	0.9467	
DEAP	0.7100	0.9861	1.9800	0.9790	2.8000	0.9782	
Wen et al. (2024a)	DEAP	0.0781	0.9999	0.0000	1.0000	0.0000	1.0000	
Graz A	0.1309	0.9985	0.0116	0.9999	7.0504	0.9237	
Bonn	0.0178	0.9998	0.0033	1.0000	0.0000	1.0000	
Proposed method	DEAP	0.0000	1.0000	0.0000	1.0000	0.0018	0.9981	
Graz A	0.0001	0.9999	0.0000	1.0000	0.0014	0.9986	
Bonn	0.0001	0.9999	0.0000	1.0000	0.0008	0.9992	

The superior robustness exhibited by the proposed SWT-SVD-tent map method primarily stems from three key mathematical characteristics, each contributing significantly to its resistance against common signal processing attacks. Firstly, SWT preserves signal length and ensures shift-invariance, making the embedding resilient to typical signal-processing attacks such as random cropping and shifting. Secondly, the SVD-based embedding into off-diagonal singular values reduces distortion of dominant signal features, maintaining high structural integrity under conditions like noise addition and filtering. Finally, the tent map’s chaotic encryption exhibits strong sensitivity to initial conditions and produces highly uniform chaotic sequences, significantly complicating attempts at unauthorized extraction or tampering. The combination of these features results in a mathematically robust embedding scheme capable of superior performance compared to alternative methods.

Comparative analysis of computational complexity

Time and space complexities constitute crucial metrics for evaluating the computational efficiency and practicality of steganographic algorithms. To assess the complexity of the proposed method, a comparative analysis was performed against the approaches presented by Akgul et al. (2020), Pham, Tran & Ma (2015), Gupta, Chakraborty & Gupta (2021), Nguyen, Pham & Nguyen (2020), and Wen et al. (2024a). Since explicit time and space complexity measures were not provided in these prior studies, approximate complexity estimates were inferred based on their described methodologies. The summarized results are given in Table 11. Gupta, Chakraborty & Gupta (2021) first applied the short-time Fourier transform (STFT) to the EEG signal, converting it into a two-dimensional representation, and then performed a two-dimensional discrete cosine transform (DCT) for information hiding. Their method exhibited a time complexity of O(n2logn) and space complexity of O(n2). By contrast, Akgul et al. (2020) utilized a combination of least significant bit (LSB) embedding and chaotic mapping. Since these operations are essentially linear, their algorithm achieved time and space complexities of O(n). The methods of Nguyen, Pham & Nguyen (2020), Pham, Tran & Ma (2015), and the proposed approach all employed SVD in their embedding processes. Pham, Tran & Ma (2015) converted the one-dimensional EEG signal into an nxn image, resulting in a time complexity of O(n3) and a space complexity of O(n2). In contrast, both Nguyen, Pham & Nguyen (2020) and the proposed method transformed the EEG signal into a nxn matrix, which reduced computational demands. Under these conditions, both approaches achieved a time complexity of O(n32) and a space complexity of O(n). Wen et al. (2024a) also employed a Wavelet Packet Transform (WPT) followed by SVD on a nxn matrix, resulting in similar computational complexities of O(n32) for time and O(n) for space. In summary, the proposed method exhibits computational complexities comparable to those of Nguyen, Pham & Nguyen (2020), and Wen et al. (2024a), which are relatively lower than the complexities associated with converting the signal into an nxn image, as seen in Pham, Tran & Ma (2015). Consequently, the proposed method remains a highly competitive and computationally efficient approach for embedding private information into EEG signals, offering advantages in practical real-time scenarios.

Table 11 Results of the comparative analysis of algorithm complexity.

Method	Time complexity	Space complexity	
Akgul et al. (2020)	O(n)	O(n)	
Pham, Tran & Ma (2015)	O(n3)	O(n2)	
Gupta, Chakraborty & Gupta (2021)	O(n2logn)	O(n2)	
Nguyen, Pham & Nguyen (2020)	O(n32)	O(n)	
Wen et al. (2024a)	O(n32)	O(n)	
Proposed method	O(n32)	O(n)	

To further analyze the efficiency of the proposed method, a comparison was conducted against (Wen et al., 2024a), examining time complexity, space complexity, and actual execution times—including hiding, extraction, and total processing durations. The summarized results are presented in Table 12. Both the proposed method and Wen et al.’s (2024a) approach exhibit comparable time complexity O(n32) and space complexity O(n), indicating a similar level of computational efficiency from a theoretical standpoint. However, their execution times differ notably. Specifically, the proposed method achieves a hiding time of 0.000258 s, which is considerably faster than Wen et al.’s (2024a) 0.001090 s. Moreover, the extraction time for the proposed method is 0.001356 s, outperforming Wen et al.’s (2024a) extraction time of 0.001710 s. Consequently, the total execution time of the proposed method is 0.001968 s, significantly lower than the 0.002800 s reported by Wen et al. (2024a). Since execution time data from other comparable studies are not available, direct comparisons were limited to Wen et al.’s (2024a) method. In conclusion, while offering similar theoretical time and space complexities, the proposed method demonstrates a distinct advantage in execution speed, making it highly suitable for real-time applications where rapid performance is essential.

Table 12 Comparison of hiding time (s) and extraction time (s).

Method	Time complexity	Space complexity	Hiding time (s)	Extraction time (s)	Total time (s)	
Proposed method	O(n32)	O(n)	0.000258	0.001356	0.001968	
Wen et al. (2024a)	O(n32)	O(n)	0.001090	0.001710	0.002800	

Practical feasibility considerations

While the primary objective of this research was to design and validate an effective EEG signal steganography algorithm, practical feasibility considerations were also thoroughly examined. The proposed method was evaluated on a system equipped with an Intel Xeon E5-2630 processor and 12 GB RAM, demonstrating low computational cost and fast execution times suitable for real-time applications. Moreover, the storage requirement of the proposed method remains minimal since only private information of limited size (up to 200 bytes) is embedded per EEG segment. The method operates on a segment-by-segment basis without altering the overall signal length, ensuring compatibility with storage and transmission systems used in telemedicine platforms. From a power consumption viewpoint, the use of lightweight encryption (tent map) combined with efficient signal processing methods (SWT and SVD) ensures compatibility with low-power devices typically employed in wearable or portable EEG monitoring systems.

Conclusions

In this study, a robust and computationally efficient method for embedding private information into EEG signals was developed by integrating the stationary wavelet transform, singular value decomposition, and tent map approaches. The primary objectives were to achieve high embedding capacity, preserve the perceptual integrity of the EEG signals, ensure resilience against various forms of interference, and maintain low computational overhead.

Comprehensive evaluations were conducted on multiple publicly available EEG datasets, employing a range of widely recognized metrics such as PSNR, SSIM, PRD, NCC, BER, and ED. The results demonstrated consistently high PSNR values, indicating that the integrity and perceptual quality of the original EEG signals were maintained. Moreover, the method displayed strong robustness against common distortions, including noise interference, random cropping, and low-pass filtering, underscoring its suitability for real-world scenarios where signal degradation is inevitable. In addition to its resilience, the proposed method exhibited favorable computational efficiency. It achieved shorter execution times for both the hiding and extraction of private information compared to conventional approaches. This efficiency is particularly advantageous for real-time or resource-constrained applications, where rapid processing and timely data recovery are essential.

Overall, this method provides a well-rounded solution that balances performance, robustness, and computational complexity, making it a viable candidate for secure and efficient information embedding in EEG signals. Future work will focus on further enhancing resistance to filtering operations, refining parameter selection, and extending the applicability of the approach to other types of biomedical signals, thereby broadening the range of its potential applications.

Supplemental Information

Supplemental Information 1 Code.

Additional Information and Declarations

Competing Interests

The authors declare that they have no competing interests.

Author Contributions

Enes Efe conceived and designed the experiments, performed the experiments, analyzed the data, performed the computation work, prepared figures and/or tables, authored or reviewed drafts of the article, and approved the final draft.

Data Availability

The following information was supplied regarding data availability:

The code is available in the Supplemental File.

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
