# Peer review of "High-fidelity steganography in EEG signals using advanced transform-based methods"

_PeerJ Computer Science, doi:10.7717/peerj-cs.2900_

## Round 0.1 · original submission · Major Revisions

Please address the reviewer comments in detail.

Reviewer 1 ·

Basic reporting

No comments

Experimental design

Data hiding process should be added with results of sample images. Same for data extraction.

Validity of the findings

Validate the findings by adding stego images and extracted images.

Cite this review as

Reviewer 2 ·

Basic reporting

see additional comment

Experimental design

see additional comment

Validity of the findings

see additional comment

Additional comments

This paper proposes steganography in EEG signals as a data protection method, but the reason for choosing the object (EEG) has not been clarified sufficiently. Before discussing the problems in the encryption method, it is necessary to explain what are the main problems in EEG data protection that cannot be solved by encryption alone. Are there any specific risks in EEG transmission or storage that make steganography a superior solution? If risks such as theft, manipulation, and detection still apply to steganography, then a deeper analysis is needed on whether steganography really reduces these risks significantly or just adds a layer of security without strong justification.

In addition, the purpose of implementing steganography in EEG has not been explicitly explained. Is it used to hide medical metadata, patient identity, or other information? Without this explanation, it is difficult to assess the urgency and practical benefits of the proposed method. It is better to strengthen the introduction with a clearer problem identification based on real needs in EEG signal processing, rather than just comparing steganography with encryption without specific context.

Why still use Tent Map, not the more modern and complex chaos method?

Is there a strong justification that Tent Map is superior in security or efficiency to the latest alternatives?

If encryption is considered vulnerable, why is it still used in this method?

Does the combination of encryption + steganography really improve security or just add complexity?

What are the main problems in EEG protection that cannot be solved by encryption alone?

What information is hidden in the EEG signal, and what are its application scenarios?

Currently, the paper uses a combination of SWT, SVD, and Tent Map, all of which are standard methods and have been widely used in steganography and signal security. What is the main novelty of this approach?

If the innovation lies in the combination of techniques, how does this method differ significantly from previous studies that also combined wavelet transform and chaos methods?

Are there any new aspects in the embedding algorithm or parameter optimization that give it an advantage over previous methods?

Without clear novelty, this method looks more like a variation of an existing approach than a truly new contribution.

Comparison with similar methods needs to be explained (Table 10), whether using the same payload, when compared with the results of Wen et al, it appears that the PSNR and MSE values ​​are not aligned. In fact, the PSNR calculation is based on MSE. Comparison should be done with the same payload. SSIM and PSNR is more recomended

The comparison table should be separated between embedding and extraction metrics.

How to compare embedding and extraction speeds? Are you using the same hardware? Or the author replicates the method needs to be explained.

Cite this review as

Reviewer 3 ·

Basic reporting

Dear Authors:
Address the comments:
1. The study lacks a detailed explanation of the Tent Map's impact on security and robustness compared to other chaotic maps such as the Logistic Map. A comparative analysis with different chaotic maps should be included to justify its selection.

2. The robustness analysis against attacks such as low-pass filtering and noise addition is insufficient. More rigorous testing with different levels of noise and real-world transmission conditions should be conducted.

3. The embedding capacity analysis does not consider the trade-off between payload size and signal integrity. A systematic study showing the effect of increasing the embedded data size on PSNR and BER is necessary.

4. The computational complexity section lacks a clear comparison of execution times with alternative methods. A direct benchmark against previously proposed EEG steganography techniques should be provided to demonstrate efficiency improvements.

5. The paper does not discuss the real-world feasibility of deploying this method in practical EEG-based applications. An analysis of hardware compatibility, storage constraints, and power consumption in real-time systems should be included.

6. The statistical validation of results is missing, with no mention of confidence intervals or standard deviations for key performance metrics. Variability across different datasets should be analyzed to ensure the method's reliability.

7. The impact of channel selection on steganographic performance is not discussed. EEG channels have different noise levels and signal characteristics; an analysis of channel sensitivity should be included to optimize embedding locations.

Experimental design

-

Validity of the findings

-

Additional comments

-

Cite this review as

Reviewer 4 ·

Basic reporting

The manuscript presents a robust and efficient steganographic method for embedding private information into EEG signals. By leveraging Stationary Wavelet Transform (SWT), Singular Value Decomposition (SVD), and Tent Map techniques, the study demonstrates strong performance in preserving signal integrity, robustness against attacks, and computational efficiency. The methodology is well-structured, and experimental results confirm the effectiveness of the proposed approach. However, there are a few areas that require clarifications, additional discussion, and minor improvements to strengthen the manuscript.

Strengths:
1. The proposed method effectively integrates SWT, SVD, and Tent Map techniques to enhance the embedding capacity while ensuring signal fidelity.
2. Performance is validated using three well-known EEG datasets (Graz A, DEAP, Bonn) and a variety of quality metrics (PSNR, MSE, PRD, NCC, BER, Euclidean Distance).
3. The method demonstrates high resilience against noise addition, random cropping, and low-pass filtering, proving its practical applicability.
4. The approach is benchmarked against existing EEG-based steganographic techniques, showcasing its advantages in embedding capacity, security, and computational efficiency.
5. The logical flow from introduction to experimental validation makes the paper easy to follow.

Major Revision Comments
• While the manuscript discusses previous EEG-based steganographic techniques, the differentiation from existing DWT-SVD and chaotic encryption methods should be made clearer.
• Add a dedicated section (or a clearer subsection) comparing the proposed approach to other hybrid steganography techniques in EEG signals.
• There are some good references about ECG Signal and optimization methods, consider the bellow refs

1. Pan, H., Wang, Y., Li, Z., Chu, X., Teng, B.,... Gao, H. (2024). A Complete Scheme for Multi-Character Classification Using EEG Signals From Speech Imagery. IEEE Transactions on Biomedical Engineering, 71(8), 2454-2462. doi: 10.1109/TBME.2024.3376603
2. Pan, H., Li, Z., Fu, Y., Qin, X., & Hu, J. (2024). Reconstructing Visual Stimulus Representation From EEG Signals Based on Deep Visual Representation Model. IEEE Transactions on Human-Machine Systems, 54(6), 711-722. doi: 10.1109/THMS.2024.3407875
3. Pan, H., Tong, S., Song, H., & Chu, X. (2025). A Miner Mental State Evaluation Scheme With Decision Level Fusion Based on Multidomain EEG Information. IEEE Transactions on Human-Machine Systems, 1-11. doi: 10.1109/THMS.2025.3538162
4. Chenhao Bao Xin Hu Dan Zhang Zhao Lv Jingjing Chen. . Predicting Moral Elevation Conveyed in Danmaku Comments Using EEGs. Cyborg Bionic Syst. 2023:4;0028. DOI:10.34133/cbsystems.0028
5. Ye W, Wang J, Chen L, Dai L, Sun Z, Liang Z. Adaptive Spatial–Temporal Aware Graph Learning for EEG-Based Emotion Recognition. Cyborg Bionic Syst. 2024;5:Article 0088. https://doi.org/10.34133/cbsystems.0088
6. Wen H, Zhong Y, Yao L, Wang Y. Neural Correlates of Motor/Tactile Imagery and Tactile Sensation in a BCI paradigm: A High-Density EEG Source Imaging Study. Cyborg Bionic Syst. 2024;5:Article0118. https://doi.org/10.34133/cbsystems.0118

• Clarify how using the Tent Map specifically enhances security compared to traditional chaotic maps like the Logistic Map.
• The manuscript discusses time and space complexities, but real-world applicability in resource-constrained environments (e.g., embedded medical devices) is not sufficiently addressed.
• Discuss the feasibility of deploying this method in real-time EEG processing scenarios such as brain-computer interfaces (BCIs) or telemedicine systems.

Minor Revision Comments
• Some sentences could be refined for better readability and flow.
• Example:
o Current: "Encryption renders the data unintelligible, preventing unauthorized access and thus securing data transmission."
o Suggested: "Encryption ensures data confidentiality by making it unintelligible to unauthorized parties, thereby securing transmission."
• A final round of professional proofreading is recommended.
• Some figures (e.g., bifurcation diagrams of Tent and Logistic Maps) could be enhanced with clearer legends and axis labels.
• The PSNR, BER, and NCC tables could include confidence intervals to strengthen the statistical analysis.
• The robustness section could be expanded to include why the method performs well against attacks.
• Include a few lines explaining the mathematical reasons why SWT-SVD-Tent Map outperforms other techniques in robustness tests.

Experimental design

No comment

Validity of the findings

No comment

Additional comments

No comment

Cite this review as

---

## Round 0.2 · accepted · Accept

Since the comments have been addressed, we are happy to inform you that your manuscript has been accepted.

Reviewer 1 ·

Basic reporting

Clear and unambiguous language

Experimental design

Research is elaborated in satisfactory manner and within the scope of the journal.

Validity of the findings

Satisfactory results

Cite this review as

Reviewer 4 ·

Basic reporting

I have carefully reviewed the revised manuscript and commend the authors for their thorough and thoughtful responses to all the comments raised in my previous review. The revisions have significantly improved the clarity, technical depth, and overall quality of the paper. All suggested corrections and recommendations have been adequately addressed.

Based on the current version of the manuscript, I am satisfied with the improvements made and have no further concerns. I therefore recommend the paper for acceptance and subsequent publication in your esteemed journal.

Experimental design

No Comment

Validity of the findings

No Comment

Additional comments

No Comment

Cite this review as